# Low Plasma Carnosinase-1 Activity in Patients with Left Ventricular Systolic Dysfunction: Implications for Carnosine Therapy in Heart Failure

**DOI:** 10.3390/ijms26062608

**Published:** 2025-03-14

**Authors:** I-Chau Liang, Ettore Gilardoni, Islam A. Berdaweel, Knute D. Carter, Ethan J. Anderson

**Affiliations:** 1Department of Pharmaceutical Sciences and Experimental Therapeutics, College of Pharmacy, University of Iowa, Iowa City, IA 52242, USA; i-chau-liang@uiowa.edu (I.-C.L.); i.berdaweel@yu.edu.jo (I.A.B.); 2Department of Clinical Pharmacy and Pharmacy Practice, Yarmouk University, Irbid 21163, Jordan; 3Department of Biostatistics, College of Public Health, University of Iowa, Iowa City, IA 52242, USA; knute-carter@uiowa.edu; 4Fraternal Order of Eagles Diabetes Research Center, University of Iowa, Iowa City, IA 52242, USA; 5Abboud Cardiovascular Research Center, University of Iowa, Iowa City, IA 52242, USA

**Keywords:** carnosinase, CNDP1, CN1 protein, carnosine, heart failure

## Abstract

Therapeutic efficacy of histidyl dipeptides such as carnosine is hampered by circulating carnosinase-1 (CN1), which catalyzes carnosine’s hydrolysis and degradation. Prior reports suggest that oral carnosine may improve cardiometabolic parameters in patients with heart failure (HF), but whether CN1 activity is affected by HF is unknown. Here, we measured CN1 content and carnosine degradation rate (CDR) in preoperative plasma samples from a cohort of patients (*n* = 138) undergoing elective cardiac surgery to determine whether plasma CN1 and/or CDR varied with left ventricular (LV) systolic dysfunction. CN1 content was normally distributed in the cohort, but plasma CDR displayed a quasi-bimodal distribution into high- (>2 nmol/(h*μL)) and low-activity (≤2 nmol/(h*μL)) clusters. Multivariable analysis confirmed female sex, diabetes and LV systolic dysfunction was associated with the low-activity CDR cluster. Although CN1 content did not differ, logistic regression analysis revealed that CDR and CN1-specific activity (CDR/CN1 content) was significantly lower in patients with both moderate (ejection fraction, EF ≥ 35 to <50%) and severe LV systolic dysfunction (EF < 35%) compared with patients in the normal range (EF ≥ 50%). These findings suggest that plasma CN1 activity is regulated by factors independent of expression, and that a decline in LV systolic function is associated with low CN1 activity. Further studies are needed to delineate specific mechanisms controlling CN1 expression and activity, which will facilitate the development of carnosine and other histidyl dipeptide therapies for cardiometabolic disorders such as HF.

## 1. Introduction

Reactive carbonyl species (RCS) generated from the breakdown of oxidized lipids (e.g., 4-hydroxynonenal (4-HNE) and malondialdehyde) and glucose (e.g., methylglyoxal) underlie the pathogenesis of many chronic diseases, particularly cardiovascular disease (CVD). The pathogenic effects of RCS are due to their nucleophilic attack on macromolecules, including proteins, DNA, phospholipids and more. A large and growing body of evidence links RCS to CVD etiology, including atherosclerosis, vascular diseases, fibrosis, obesity and diabetes [1,2,3,4,5,6,7]. Importantly, high levels of RCS, particularly 4-HNE, have been associated with ventricular hypertrophy and dysfunction in patients with heart failure (HF) [4,5,8]. For these reasons, pharmacotherapies that specifically neutralize RCS would be advantageous for preventing and treating CVD.

Histidine-containing dipeptides such as carnosine and anserine are potent scavengers of RCS and have shown therapeutic potential in preclinical studies and small clinical trials for cardiometabolic diseases. Carnosine, a β-alanyl histidine dipeptide that exists in micro-to-millimolar concentrations in heart, brain and skeletal muscle, has been the most studied in the context of CVD therapy. Oral carnosine has shown promising effects in mitigating dyslipidemia [9] and hyperglycemia [10] in patients with metabolic syndrome, and improving exercise performance in patients [11] and rodent models of cardiomyopathy [12,13,14]. However, there are significant challenges associated with oral carnosine as a pharmacotherapy in humans; first and foremost is the presence of carnosinase-1 (CN1) in the bloodstream, which hydrolyses carnosine and thus negates its ability to reduce RCS.

Carnosinase was first identified through experiments on swine kidney in 1949, where it was determined that the enzyme hydrolyzes carnosine following a zero-order kinetic reaction and that its activity can be inhibited by metal poisoning [15]. CN1 and CN2 are the two known isoforms in mammals, differing by their metal ion interaction and their tissue localization. In higher order mammals (not rodents), CN1 contains a N-terminal signal peptide which enables it to be secreted, while CN2 is intracellularly localized. Molecular studies using human tissues have shown that while CN2 is ubiquitously expressed in all tissues, CN1 is almost exclusively expressed in brain, suggesting that this tissue is, at the very least, a major source of circulating CN1 [16,17,18]. CN1 activity has been reported to be higher in females than males and it increases with age up to 40 years when its activity reaches a plateau [18,19]. Recent genetic and epidemiological studies have linked CNs to numerous human diseases, such as neurodegeneration, muscle disorders, ageing and cardiometabolic disorders [18,20,21,22,23,24,25,26,27,28]. Despite these reports, the link between CN1 content and activity is not always clear, with most reports documenting either CN1 content alone, or plasma/serum carnosine degradation rate (CDR) as an index of CN1 activity. Furthermore, the directionality of the associations between CN1 content/activity and disease is not always similar. Clinical case reports have linked CN1 deficiency to progressive neurological disorders such as motor dysfunction and intellectual disability in children, and Parkinson’s disease and multiple sclerosis in adults [29,30,31]. Additionally, chronic liver disease was associated with decreased serum CN1 activity in adult patients [32]. Conversely, CN1 deficiency has been associated with positive clinical outcomes in other studies. Studies in adult patients with type 2 diabetes mellitus (T2DM) have shown that low serum CN1 concentration is associated with significantly reduced risk for diabetic nephropathy, with the strongest effect seen in female patients. The beneficial effect of low CN1 on diabetic nephropathy was ascribed to the higher levels of carnosine consequently seen in the patients as a result of the lower CN1 levels [23,24,25]. Taken together, it is clear from these findings that many biological and pathological variables can influence the expression level and activity of CN1. Moreover, the contradictory findings among the various clinical reports of CN1 strongly indicate that its expression and activity may be differentially impacted by disease type.

As discussed above, experimental and clinical studies have suggested carnosine has a beneficial effect in HF patients. Since carnosine’s therapeutic potential is heavily influenced by CN1, it is important to know the extent to which CN1 expression level and activity are changing with myocardial dysfunction. The objective of the present study was to determine whether plasma CN1 content (i.e., expression level) or activity was altered in patients with moderate or severe LV systolic dysfunction compared with normal function in a cohort of patients undergoing elective coronary artery bypass graft (CABG) surgery.

## 2. Results

### 2.1. Clinical and Demographic Variables

Preoperative plasma samples were collected from adult patients undergoing primary elective coronary artery bypass graft (CABG) or CABG + valve repair/replace surgery from 2014 to 2018. A total of *n* = 138 patients were ultimately used for this analysis based on inclusion/exclusion criteria (see Section 4.1 below) and the availability of matching clinical data (see Figure 1). Major clinical and demographic variables of the patient cohort are shown in Table 1 and categorized into three groups based on LV systolic functional status prior to being admitted for surgery—normal (ejection fraction (EF) ≥ 50%), moderate (EF ≥ 35% to <50%) and severe (EF < 35%) LV systolic dysfunction. Of the entire cohort, 17.4% and 19.6% had moderate and severe LV systolic dysfunction, respectively. Clinical factors associated with severe LV systolic dysfunction were sex (*p* < 0.01), race (*p* = 0.02), prior MI (*p* = 0.05) and diagnosis of HF (*p* < 0.01). Patients were otherwise well matched for age, comorbidities and preoperative medications across the three groups.

### 2.2. Cardiac Function

Echocardiographic parameters of cardiac function among patients in each LV systolic function group are shown in Table 2, with significant differences seen in fractional shortening (FS), left ventricular (LV) mass, end-diastolic (EDV) and -systolic volume (ESV), stroke volume (SV) and mitral valve E/A ratio, with the differences of the greatest magnitude between normal (EF ≥ 50%) and severe (EF < 35%) LV systolic dysfunction groups (*p* < 0.01, Kruskal–Wallis).

### 2.3. Univariable Analysis of Serum Carnosinase Content and Activity

Plasma CN1 concentration (Figure 2A) was not significantly different across LV functional status groups (*p* = 0.34). CDR was clustered in a quasi-bimodal distribution into ‘high-’ and ‘low carnosine metabolizer’ groups (Figure 2B). Patients with severe LV systolic dysfunction (EF < 35%) had substantially lower CDR compared with patients who had normal LV systolic function n (CDR = 1.95 ± 1.7 nmol/h*μL vs. 2.93 ± 1.6 nmol/h*μL, respectively, *p* = 0.02). This corresponded to differences in CN1-specific activity (Figure 2C) between these groups that did not quite reach statistical significance (193.5 ± 331.8 nmol/ng*h vs. 368.7 ± 408.7 nmol/ng*h, respectively, *p* = 0.11).

### 2.4. Multivariable Analysis of Serum Carnosinase Content and Activity

Multivariable logistic models were constructed to identify characteristics associated with ‘low vs. high carnosine metabolizers’, which were discriminated using the following cutoff values: CDR = 2 nmol/(h*μL), CN1 = 80 ng/mL, and CN1-specific activity = 200 nmol/(ng*h) (see dotted lines in the panels of Figure 2 above). Logistic regression analysis of CDR (Table 3) revealed that severe LV systolic dysfunction (EF < 35%) was associated with a 5.93-fold increase in the odds ratio (OR) for having patients with CDR < 2 nmol/(h*μL) compared with normal LV systolic function. Males were significantly more likely than females to have CDR > 2 nmol/(h*μL) (OR = 5.89 vs. females). Additionally, having a prior diagnosis of DM was associated with CDR < 2 nmol/(h*μL) (OR = 0.46).

Multivariable analysis further revealed that after adjusting for DM and sex, CN1-specific activity was significantly lower in patients with both moderate (EF ≥ 35% to <50%) and severe LV systolic dysfunction (EF < 35%) compared with patients with normal LV systolic function (Table 4).

## 3. Discussion

Genetic and epidemiological reports concerning the effects of disease on CN1 have been difficult to reconcile because CN1 expression and activity are not frequently measured concurrently, and in the few cases where they are, results have been variable, conflicting and often differ by disease type. The main objective of the present study was to determine whether carnosine degradation and CN1 levels in plasma were associated with a decline in myocardial function. No studies to date have sufficiently addressed this question, and this is an important knowledge gap given the promising effects seen with oral carnosine supplementation in patients with cardiometabolic disease, including HF. We found that plasma CN1 enzyme levels were not significantly changed in patients with LV systolic dysfunction. The more interesting finding pertained to plasma CDR and CN1-specific activity. First, CDR in this cohort was not normally distributed among these patients and displayed a quasi-bimodal clustering into ‘high’ and ‘low carnosine metabolizer’ groups. Similar to prior reports, we found a weak but significant association between DM and low CN1 activity (Table 3), but the greatest differences were found to be in patients with severe LV systolic dysfunction (EF < 35%), which was strongly associated with lower CDR and CN1-specific activity (i.e., low carnosine metabolizers; see Figure 2 and Table 3). Together these results suggest that factors independent of CN1 expression are responsible for regulating its activity, and these factors are influenced either directly or indirectly by cardiometabolic health.

One of the strengths of our study is the racial diversity of the patient cohort, which comprised ~22% African Americans. There were more African American patients in the severe LV systolic dysfunction group (EF < 35%) compared with the other groups, and this is in line with the known higher risk of HF in patients of African descent [33,34]. Interestingly, several genetic epidemiology studies have demonstrated that CN1 secretion is influenced by CTG polymorphism (i.e., leucine repeat) in the *CNDP1* gene, and this polymorphism is associated with race and ethnicity [35,36,37]. Genotyping studies in African American patients showed that loci near the carnosinase genes (*CNDP1* and *CNDP2*) in chr18 contained a number of variants associated with diabetic end-stage renal disease, suggesting that these variants may contribute in part to increased susceptibility to diabetic nephropathy in African Americans [38,39]. The effect of these variants on carnosinase expression and activity is not at all clear, however, as there have not been comprehensive genotype–phenotype studies performed in large mixed-race/ethnicity cohorts. One study reported a borderline significant association between the *CNDP1* genotype and serum CN1 concentration, though the association with CN1 activity did not hold [27]. In any case, we cannot exclude the possibility that ethnicity-related genotypic differences in our patient cohort may partially explain the existence of the ‘low and high carnosine metabolizers.’ However, it is not likely that the slight differences in number of African American patients in the severe LV systolic dysfunction group explains the lower CN1-specific activity seen in these patients since the total percentage of African American patients in each of the LV function groups is otherwise well matched.

Carnosinase is known to exist in both monomeric and homo-dimeric forms in vivo; the latter comprises two subunits connected by one or more disulfide bonds [17,40]. The dimeric form provides more accessible hydrogen bonds for carnosine at the substrate binding site, which is thought to underlie the higher activity in this form compared with the monomer [40]. Although it is intriguing to consider that differences in the ratio between monomer and dimer forms of the enzyme may be contributing to the proportion of ‘low’ and ‘high carnosine metabolizers’ within patients, the mechanisms underlying the macromolecular configuration of CN1 remain unknown. It is possible that *CNDP1* genotype, disease, medications and potentially many other stimuli might be differentially impacting the monomer/dimer ratios in this cohort. Unfortunately, the commercial assay we used to measure CN1 content in the present study contains an antibody that cannot distinguish between monomer and dimeric forms, which would require more extensive biochemical analysis.

Early studies revealed that CN1 activity is highly sensitive to metal ions, particularly cadmium (Cd^2+^). Although it is not required for activity, high concentrations (≥100 µM) of Cd^2+^ greatly enhance CN1 activity and low concentrations (<3 µM) diminish its activity [41]. Smoking is well known to cause increases in plasma Cd^2+^ levels [42,43]. Most of the patients in this study had some history of tobacco use, although the ratio of active compared with former smokers may have been variable across the groups. Thus, we cannot exclude the possibility that tobacco use or smoking is an additional variable that may be driving the differences in CDR seen in our patient cohort.

Since bloodborne CN1 levels are tightly linked to CN1 levels in cerebrospinal fluids, serum carnosinase has an extensive history as a clinical biomarker for neurological disorders and injury. Decreased serum CN1 concentrations have been linked with neurodegenerative diseases such as Parkinsons’ and multiple sclerosis [31] and as a biomarker predicting poor outcomes following stroke [44]. Two reports in patients undergoing cardiopulmonary bypass (CPB) surgery revealed that serum carnosinase decreases as CPB time increases, with the implication that circulating CN1 levels may be indicative of brain ischemia as a result of CPB [45,46]. One very interesting recent study in a cohort of T2 diabetes patients showed that serum CN1 predicted renal function impairment over a follow-up period of ~1.5 years [26]. Although we excluded patients with albuminuria from the present study, it is still possible that some variability in renal function among the patients in the present study may partly explain some of the differences in CN1 levels seen across the LV systolic function groups.

Limitations of our study include the relatively small sample size and single-institution enrollment. Additional biochemical analysis is clearly needed to delineate the mechanisms explaining the ‘low’ and ‘high carnosine metabolizer’ populations given the CDR distribution observed in this study. The potential contribution of other serum proteases on CDR cannot be excluded, though it is unlikely to be substantial given that no other serum dipeptidase has been reported to have the rapid kinetic profile for carnosine as CN1 [47].

Taken together, our findings indicate that patients with LV systolic dysfunction have lower CDR compared with patients who have normal LV function, suggesting that patients with HF (specifically HF with reduced ejection fraction, HFrEF) may benefit from oral carnosine supplementation. Further studies of plasma CN1 concentration and activity in larger patient cohorts with comprehensive matching clinical data and longitudinal follow-up are necessary to provide more robust conclusions on the association of this enzyme as it relates to HF.

## 4. Materials and Methods

### 4.1. Study Design and Plasma Sample Collection

The Institutional Review Boards of East Carolina University and the University of Iowa reviewed and approved all aspects of this study. A total of 138 patients undergoing on-pump elective coronary artery bypass graft (CABG) or CABG and valve repair/replace surgery at the East Carolina Heart Institute between August 2014 and December 2018 were included in this analysis. Elderly patients (>75 years), those undergoing maze procedures (surgical or catheter), and those with preoperative shock, severe renal dysfunction (i.e., albuminuria) and a body mass index (BMI) of >35 kg/m^2^ were excluded. Patients were stratified into groups according to LV systolic functional status as indicated by echocardiography at time of preoperative evaluation.

A preoperative fasting blood sample was obtained from each patient on the morning of surgery prior to anesthesia (Figure 1) using sodium citrate vacutainer tubes and transferred on ice to the laboratory within 30 min. Samples were centrifuged for 10 min at 4000 rpm; plasma and buffy coat were transferred to another tube and further centrifuged at 8000 rpm for 10 min. The remaining plasma was divided into cryotube aliquots for storage at −80 °C.

### 4.2. Plasma Carnosine Degradtion Rate (CDR) and CN1 Concentration Measurements

Plasma CDR measurements were obtained using a slightly modified LC-MS method described previously [48]. In brief, 80 µM carnosine was incubated in plasma previously warmed at 37 °C for 5 min and mixed with 10 mM of phosphate-buffered saline (pH 7.45) to give a total volume of 300 μL. Reactions were terminated at t = 0 min (immediately) and after t = 5 min by adding trichloroacetic acid (TCA). After centrifuging at 10,000 rpm for 10 min at room temperature, the supernatants were diluted by a ratio of 1:10 with 0.1% formic acid (FA) and subjected to LC-MS analysis with an Accurate-Mass 6530 Q-TOF mass spectrometer (Agilent Technologies, Santa Clara, CA, USA). CDR was then calculated as previously described [48].

Serum CN1 concentration was determined using a commercial human CNDP1 ELISA kit ELH-CNDP1 (RayBiotech, Peachtree Corners, GA, USA). The capture antibody was a mouse monoclonal IgG1 antibody generated against amino acids 28–507 of recombinant human CNDP1 (i.e., CN1). The detection antibody was a polyclonal goat IgG with the same immunogen. The antigen standard provided was an NS0-derived recombinant human CNDP1 with an expression region of aa 28–507 with a c-terminal 10-His tag. All measurements of the patient’s sample were executed according to the manufacturer’s instructions, and final absorbance readings (450 nm) were made using a BioTek Epoch microplate spectrophotometer (Agilent Technologies, Santa Clara, CA, USA). The minimum detectable concentration of human CN1 was determined to be 1.3 ng/mL. Each run contained duplicate standards and samples. The intra-assay covariance percentage (CV%) was determined to be <10% and the inter-assay CV% was <12%.

### 4.3. Statistical Analysis

Univariable analysis of patient baseline characteristics and clinical data was performed using chi-square tests for categorical variables; Fisher’s Exact Test was used if the expected cell counts were too small. Patients were separated into three groups based on I/E criteria described above, and baseline characteristics were compared across the three groups. Due to non-normality of continuous variables, the Kruskal–Wallis Test was used for comparisons across groups. Logistic regression was used for multivariable modelling of dichotomized continuous variables. A *p*-value of less than 0.05 was selected for significance. All statistical analyses were computed using SAS 9.4, and graphs were generated via GraphPad Prism version 10.4.

## 5. Conclusions

Oral carnosine supplementation has shown therapeutic potential for patients with cardiometabolic disease, including HF, but its hydrolysis via bloodborne CN1 severely limits its efficacy. This study investigated whether plasma CN1 levels and carnosine degradation rate (CDR) were associated with LV systolic dysfunction in a cohort of patients undergoing elective cardiac surgery. Although CN1 levels were not significantly different, we found that CDR and CN1-specific activity were not normally distributed and displayed a quasi-bimodal clustering into ‘high-’ and ‘low-carnosine metabolizer’ groups. Importantly, patients with severe LV systolic dysfunction (EF < 35%) had substantially lower CDR and CN1-specific activity (i.e., low carnosine metabolizers) compared with normal LV systolic function patients (EF ≥ 50%). These findings suggest that oral carnosine therapy may be beneficial in patients with HFrEF given the low rates of CN1 activity seen in this patient population. Future studies of CN1 in humans are clearly needed to delineate mechanisms underlying the disconnect between CN1 content and activity in the blood, and to confirm generalizability of these findings in other patient populations.

## Figures and Tables

**Figure 1 ijms-26-02608-f001:**
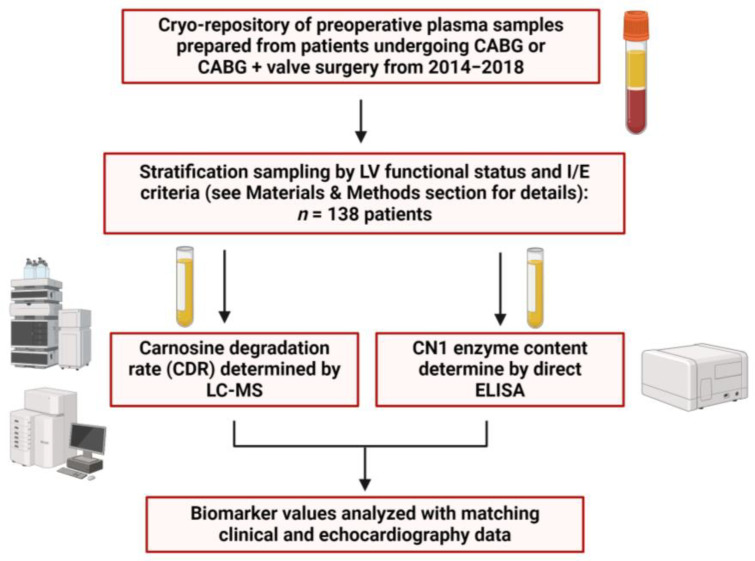
Study design. Samples of plasma were obtained from a total of 594 patients undergoing CABG only or CABG + valve surgery between 2014 and 2018. Of these, *n* = 138 samples were then used for this study based on inclusion/exclusion criteria, sample availability and/or absence of matching clinical data for each sample. CDR and CN1 concentrations were then assessed in plasma samples from each patient using liquid chromatography–mass spectrometry (LC-MS) and enzyme-linked immunosorbent assay (ELISA), respectively. Created using bioRender.com.

**Figure 2 ijms-26-02608-f002:**
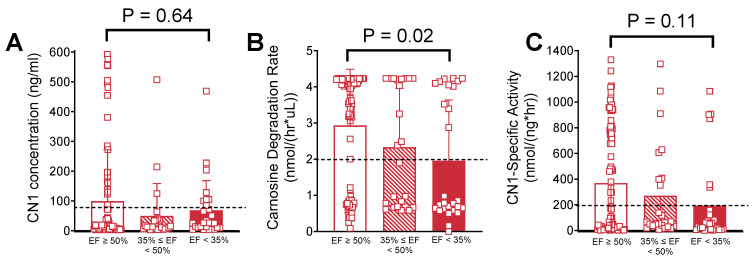
Plasma CN1 concentration; CDR and CN1 activity grouped by LV functional status. Shown in the panels above are (**A**) plasma CN1 concentration, (**B**) CDR and (**C**) CN1-specific activity in patients with normal (EF ≥ 50%), moderate (EF ≥ 35% to <50%) and severe LV systolic dysfunction (EF < 35%). The horizontal dotted line in each graph indicates the dichotomized cutoff point for statistical analysis: CN1 concentration—80 ng/mL; CDR—2 nmol/(h*μL); and CN1-specific activity—200 nmol/(ng*h).

**Table 1 ijms-26-02608-t001:** Clinical and demographic information.

Variables *n* (%)	EF ≥ 50% *n* = 87	EF ≥ 35 to <50% *n* = 24	EF < 35% *n* = 27	*p*-Value (Chi-Square)
Overall (Total *n* = 138)	87 (63.0%)	24 (17.4%)	27 (19.6%)	
**Demographic**				
Age—yr (Mean ± SD)	65 ± 6.3	64.5 ± 5.9	63.9 ± 6.8	0.80 †
Caucasian Race (%)	73 (83.9%)	20 (83.3%)	16 (59.3%)	0.02
African American	14 (16.1%)	4 (16.7%)	11 (40.7%)	
Obese (BMI ≥ 30 kg/m^2^)	31 (35.6%)	6 (25.0%)	8 (29.6%)	0.58
Male	51 (58.6%)	22 (91.7%)	21 (77.8%)	<0.01
Female	36 (41.4%)	2 (8.3%)	6 (22.2%)	
**Comorbidities**				
DM	44 (50.6%)	11 (45.8%)	15 (55.6%)	0.79
HF	6 (6.9%)	6 (25.0%)	14 (51.9%)	<0.01
History of Afib	7 (8.2%)	3 (12.5%)	2 (7.4%)	0.83 *
COPD	8 (9.3%)	4 (16.7%)	2 (7.4%)	0.56 *
Prior MI	34 (39.1%)	14 (58.3%)	17 (63.0%)	0.045
HTN	72 (82.8%)	17 (70.8%)	24 (88.9%)	0.256 *
**Medication Use**				
BB	69 (79.3%)	20 (83.3%)	19 (70.4%)	0.49
ACEI/ARB	36 (41.4%)	10 (41.7%)	10 (37.0%)	0.92
Diuretic	30 (34.5%)	4 (16.7%)	11 (40.7%)	0.16
CCB	19 (21.8%)	5 (20.8%)	2 (7.4%)	0.24
Nitrates	57 (65.5%)	18 (75.0%)	14 (51.9%)	0.21
Statin	70 (80.5%)	21 (87.5%)	22 (81.5%)	0.77 *
**Antihyperglycemic Medications**				
DM medications	39 (44.8%)	11 (45.8%)	14 (51.9%)	0.81
Insulin	27 (31.0%)	8 (33.3%)	13 (48.2%)	0.26
Metformin	18 (20.7%)	3 (12.5%)	1 (3.7%)	0.08 *
Sulfonylurea	4 (4.6%)	1 (4.2%)	1 (3.7%)	1.00 *
Thiazolidinediones	1 (1.2%)	0 (0%)	0 (0%)	1.00 *
GLP-1 agonist	2 (2.3%)	0 (0%)	0 (0%)	1.00 *
DPP-4 inhibitors	1 (1.2%)	0 (0%)	0 (0%)	1.00 *

Abbreviations: EF = ejection fraction; BMI = body mass index; DM = diabetes mellitus; HF = heart failure, Afib = atrial fibrillation; COPD = chronic obstructive pulmonary disease; MI = myocardial infarction; HTN; hypertension; BB = beta blocker; ACEI = angiotensin-converting enzyme inhibitor; ARB = angiotensin receptor blocker; CCB = calcium channel blocker; GLP-1 = glucagon-like peptide 1; DPP-4 = dipeptidyl peptidase IV; SGLT-2 = sodium–glucose transport protein 2. † Kruskal–Wallis Test; * Fisher’s Exact Test.

**Table 2 ijms-26-02608-t002:** Cardiac parameters across LV systolic functional status groups.

Variables Total *n* = 138 Mean ± SD	EF ≥ 50% *n* = 87	EF ≥ 35 to <50% *n* = 24	EF < 35% *n* = 27	*p*-Value (Kruskal–Wallis Test)
MV E/A	0.95 ± 0.36	0.94 ± 0.45	1.38 ± 0.73	0.02
FS (%)	39.2 ± 38.5	24.9 ± 8.9	16.2 ± 7.2	<0.01
LV mass DI (g)	86.8 ± 24	100.7 ± 37.5	109.7 ± 31.5	<0.01
LA dimension (cm)	3.67 ± 0.5	3.53 ± 0.6	3.73 ± 0.7	0.60
EDV (Teich) (mL)	98.8 ± 31	114.5 ± 25.6	143.3 ± 39.6	<0.01
ESV (Teich) (mL)	37.2 ± 17.8	61.3 ± 26.8	96.9 ± 36	<0.01
SV (Teich) (mL)	61.8 ± 21.6	51.8 ± 15.9	46.4 ± 22.6	<0.01

Abbreviations: EF = ejection fraction; MV E/A = mitral valve velocity (cm/s) E-wave velocity/A wave velocity ratio; FS = fractional shortening; LV = left ventricle; DI = diastole; LA = left atrial; EDV = end-diastolic volume; ESV = end-systolic volume; SV = stroke volume; Teich = Teichholz method.

**Table 3 ijms-26-02608-t003:** Logistic regression analysis of CDR.

Odds Ratio Estimates			
Variable	Ref	Point Estimate	95% Wald Confidence Limits
		**Odds of CDR > 2 nmol/(h*μL)**	
DM	No DM	0.46	(0.21, 0.99)
EF ≥ 35 to <50%	EF < 35%	1.10	(0.34, 3.56)
EF ≥ 50%	EF < 35%	5.93	(2.10, 16.79)
Male	Female	5.89	(2.34, 14.81)

Abbreviations: EF = ejection fraction; DM = diabetes mellitus. OR point estimate of 1 implies no difference.

**Table 4 ijms-26-02608-t004:** Logistic regression analysis of CN1-specific activity.

Odds Ratio Estimates			
Variable	Ref	Point Estimate	95% Wald Confidence Limits
		**Odds of CN1-Specific Activity > 200 nmol/(ng*h)**	
EF ≥ 35 to <50%	EF < 35%	1.75	(0.51, 6.06)
EF ≥ 50%	EF < 35%	3.12	(1.15, 8.48)

Abbreviations: EF = ejection fraction. OR point estimate of 1 implies no difference.

## Data Availability

All data generated and analyzed in this study are included in the article. No publicly available data sets were used for this study. Data may be available on request. Further inquiries can be directed to the corresponding author.

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
