# Peer review of "Low Plasma Carnosinase-1 Activity in Patients with Left Ventricular Systolic Dysfunction: Implications for Carnosine Therapy in Heart Failure"

_ijms, 2025, doi:10.3390/ijms26062608_

Round 1
Reviewer 1 Report
Comments and Suggestions for Authors
Low plasma carnosinase-1 activity in patients with left ventricular systolic dysfunction: Implications for carnosine therapy in heart failure
I have the following concerns:
Abstract
As stated in the ‘Instructions for Authors’ section of the IJMS journal, the abstract must not contain more than 200 words. The abstract of the following article, despite being precisely written and very explanatory, contains 273 words. I suggest revising the text.
Introduction
Page 1
“Carnosine, a β-alanyl histidine dipeptide that ex-ists in micro-to-millimolar concentrations in heart, brain and skeletal muscle, has been studied the most in the context of CVD therapy.”
Replace with: “Carnosine, a β-alanyl histidine dipeptide found in micro-millimetre concentrations in the heart, brain and skeletal muscle, has been the most studied in the context of CVD therapy.”
Page 2
“However, there are significant challenges associated with oral carnosine as a pharma-cotherapy in humans, foremost of which is the presence of carnosinase-1 (CN1) in the bloodstream which hydrolyzes carnosine and thereby abrogates its RCS-scavenging capacity.”
Replace with: “However, there are significant challenges associated with oral carnosine as pharmacotherapy in humans, first and foremost the presence of carnosine kinase-1 (CN1) in the bloodstream, which hydrolyses carnosine and thus negates its ability to reduce RCS.”
“Unlike rodents, which lack bloodborne CN1, humans have CN1 and CN2 which exist in blood and intracellular compartments, respectively.”
Replace with: “Unlike rodents, which have no CN1 in the blood, humans have CN1 and CN2 existing in the blood and intracellular compartments respectively.” Moreover, since this is the first time it has been introduced into the text, it is necessary to decline bold abbreviation in full.
Results
Page 3
2.1. Clinical and demographic variables
In the caption of Figure 1, in section 2.1 of the Results section, it is stated that plasma samples were obtained from 594 patients and only 138 were analysed for the study. I suggest that the initial number of patients involved in the study (594) should also be made explicit in the corresponding text and figure.
In the caption of Figure 1, in section 2.1 of the “Results” section, the I/E criteria are mentioned. Since this is the first time it has been introduced into the text, it is necessary to decline the bold abbreviation in full. I would also suggest clarifying what your study's inclusion and exclusion criteria were.
In the caption of Figure 1, in section 2.1 of the “Results” section, LC-MS is mentioned. Since this is the first time it has been introduced into the text, it is necessary to decline the bold abbreviation in full.
2.3. Univariable analysis of serum carnosinase content and activity
Page 5
In Figure 2, panel A, the p-value is not shown. Is this a choice?
2.4. Multivariable analysis of serum carnosinase content and activity
I suggest modifying the two tables, table 3 and table 4, by adding after the column entitled ‘Effect’ another column entitled ‘Reference Category’, for example, to make explicit which group the OR value refers to. Stating ‘EF ≥ 35 to < 50% vs EF < 35%’ or ‘Male vs Female’ on the same line of the table does not immediately make it clear who the OR value refers to.
3.Discussion
Page 6
“The more interesting finding pertained to plasma carnosinase activity, as measured and reported as CDR and CN1 specific activity (i.e., CDR/CN1 concentration).”
“Similar to prior reports, we found a weak but significant association between DM and low CN1 activity, but the greatest differences were found to be in patients with LV systolic dysfunction, where severe dysfunction (EF < 35%) was strongly associated with lower CDR and CN1 specific activity (i.e., low carnosine metabolizers).”
I suggest including the reference to tables in the text.
“One of the strengths of our study is the racial diversity of the patient cohort which is comprised of ~22% African Americans.”
I suggest including this in Table 1, of the ‘Results’ section, under the heading ‘Caucasian race’ and perhaps make it explicit in the ‘clinical and demographic variables’ section.
4.Experimental Section
I suggest changing the title of the ‘Experimental Section’ to ‘Material and Methods’ as required by the journal guidelines.

Author Response
Abstract
Comment: As stated in the ‘Instructions for Authors’ section of the IJMS journal, the abstract must not contain more than 200 words. The abstract of the following article, despite being precisely written and very explanatory, contains 273 words. I suggest revising the text.
Response: Revised abstract has been tightened up considerably and now contains 230 words. This is about as concise as it can be given that some of the CN1 variables are complicated to describe and require many characters (e.g., CDR, CN1 activity).
Introduction
Page 1
Comment: “Carnosine, a β-alanyl histidine dipeptide that ex-ists in micro-to-millimolar concentrations in heart, brain and skeletal muscle, has been studied the most in the context of CVD therapy.”
Replace with: “Carnosine, a β-alanyl histidine dipeptide found in micro-millimetre concentrations in the heart, brain and skeletal muscle, has been the most studied in the context of CVD therapy.”
Response: Manuscript text has been changed as suggested.
Page 2
Comment: “However, there are significant challenges associated with oral carnosine as a pharma-cotherapy in humans, foremost of which is the presence of carnosinase-1 (CN1) in the bloodstream which hydrolyzes carnosine and thereby abrogates its RCS-scavenging capacity.”
Replace with: “However, there are significant challenges associated with oral carnosine as pharmacotherapy in humans, first and foremost the presence of carnosine kinase-1 (CN1) in the bloodstream, which hydrolyses carnosine and thus negates its ability to reduce RCS.”
Response: Manuscript text has been changed as suggested.
Comment: “Unlike rodents, which lack bloodborne CN1, humans have CN1 and CN2 which exist in blood and intracellular compartments, respectively.”
Replace with: “Unlike rodents, which have no CN1 in the blood, humans have CN1 and CN2 existing in the blood and intracellular compartments respectively.” Moreover, since this is the first time it has been introduced into the text, it is necessary to decline bold abbreviation in full.
Response: Manuscript text has been changed as suggested.
Results
Page 3
2.1. Clinical and demographic variables
Comment: In the caption of Figure 1, in section 2.1 of the Results section, it is stated that plasma samples were obtained from 594 patients and only 138 were analysed for the study. I suggest that the initial number of patients involved in the study (594) should also be made explicit in the corresponding text and figure.
Response: Manuscript text has been changed as suggested.
Comment: In the caption of Figure 1, in section 2.1 of the “Results” section, the I/E criteria are mentioned. Since this is the first time it has been introduced into the text, it is necessary to decline the bold abbreviation in full. I would also suggest clarifying what your study's inclusion and exclusion criteria were.
Response: Thank you for this excellent suggestion. We now describe the I/E criteria in the Materials and Methods and have amended section 2.1 to refer the reader to section 4.1 where the criteria are described.
Comment: In the caption of Figure 1, in section 2.1 of the “Results” section, LC-MS is mentioned. Since this is the first time it has been introduced into the text, it is necessary to decline the bold abbreviation in full.
Response: Manuscript text has been changed as suggested.
2.3. Univariable analysis of serum carnosinase content and activity
Page 5
Comment: In Figure 2, panel A, the p-value is not shown. Is this a choice?
Response: For consistency we have now included P values above all three panels in the revised manuscript.
2.4. Multivariable analysis of serum carnosinase content and activity
Comment: I suggest modifying the two tables, table 3 and table 4, by adding after the column entitled ‘Effect’ another column entitled ‘Reference Category’, for example, to make explicit which group the OR value refers to. Stating ‘EF ≥ 35 to < 50% vs EF < 35%’ or ‘Male vs Female’ on the same line of the table does not immediately make it clear who the OR value refers to.
Response: Thank you for this helpful suggestion and to improve clarity for readers we have revised Tables 3 and 4. We have decided that using the term ‘Effect’ in those Tables is highly misleading, so we changed it to ‘Variable.’ We also added a ‘Referent’ column as suggested by the reviewer to clarify which group the OR value refers to.
3.Discussion
Page 6
“The more interesting finding pertained to plasma carnosinase activity, as measured and reported as CDR and CN1 specific activity (i.e., CDR/CN1 concentration).”
“Similar to prior reports, we found a weak but significant association between DM and low CN1 activity, but the greatest differences were found to be in patients with LV systolic dysfunction, where severe dysfunction (EF < 35%) was strongly associated with lower CDR and CN1 specific activity (i.e., low carnosine metabolizers).”
Comment: I suggest including the reference to tables in the text.
Response: Manuscript text has been changed as suggested.
Comment: “One of the strengths of our study is the racial diversity of the patient cohort which is comprised of ~22% African Americans.”
I suggest including this in Table 1, of the ‘Results’ section, under the heading ‘Caucasian race’ and perhaps make it explicit in the ‘clinical and demographic variables’ section.
Response: Table 1 of the manuscript has been changed as suggested.
4.Experimental Section
Comment: I suggest changing the title of the ‘Experimental Section’ to ‘Material and Methods’ as required by the journal guidelines.
Response: Manuscript text has been changed as suggested.

Reviewer 2 Report
Comments and Suggestions for Authors
The manuscript submitted by Liang IC et al., “Low plasma carnosinase-1 activity in patients with left ventricular systolic dysfunction: Implications for carnosine therapy in
heart failure,” determined the levels of carnosinase-1 in patients with Heart failure. This manuscript requires revision to enhance its readability for the audience.
- The authors must include information about the female patients in their study.
- The authors must establish a correlation between carnosinase-1, blood pressure, and CK-MB.
- The authors must clearly outline the specific criteria used to exclude and include participants in their study.
- The author should provide the details in the context of the patient population that experienced a second myocardial infarction. Did the authors discover any variations in carnosinase-1 levels?
- The authors must provide specific details about the number of patients in their study population who took carnosine supplements.
- The authors must include a table that illustrates the diversity of the patient population in their study.
- The authors must justify their choice of plasma samples over serum samples.
- The authors must provide data regarding CN1 levels from freshly extracted plasma versus stored plasma levels.
